# Leave Graphs Alone: Addressing Over-Squashing without Rewiring

**Domenico Tortorella**
University of Pisa
domenico.tortorella@phd.unipi.it

**Alessio Micheli**
University of Pisa
micheli@di.unipi.it

## Abstract

Recent works have investigated the role of graph bottlenecks in preventing long-range information propagation in message-passing graph neural networks, causing the so-called 'over-squashing' phenomenon. As a remedy, graph rewiring mechanisms have been proposed as preprocessing steps. Graph Echo State Networks (GESNs) are a reservoir computing model for graphs, where node embeddings are recursively computed by an untrained message-passing function. In this paper, we show that GESNs can achieve a significantly better accuracy on six heterophilic node classification tasks without altering the graph connectivity, thus suggesting a different route for addressing the over-squashing problem.

## 1 Challenges in Node Classification

Relations between entities, such as paper citations or links between web pages, can be best represented by graphs. Since the introduction of pioneering models such as *Neural Network for Graphs* [1] and *Graph Neural Network* [2], a plethora of neural models have been proposed to solve graph-, edge-, and node-level tasks [3–5], most of them sharing an architecture structured in layers that perform local aggregations of node features, e.g. graph convolution networks (GCNs) [6–8]. However, as the development of deep learning on graphs progressed, several challenges preventing the computation of effective node representations have emerged. Li et al. [9] first presented *over-smoothing* as an issue by analysing the accuracy decay as the number of layers increases in deep graph convolutional networks on semi-supervised node classification tasks. Oono and Suzuki [10] showed that repeated applications of a GCN layer cause the node representations to asymptotically converge to a low-frequency subspace of the graph spectrum. Furthermore, by acting as a low-pass filter, GCNs representation are biased in favour of tasks whose graphs present an high degree of homophily, that is nodes in the same neighbourhood share the same class [11]. In general, the inability to extract meaningful features in deeper layers for tasks that require discovering long-range relationships between nodes is called *under-reaching*. Alon and Yahav [12] maintain that one of its causes is *over-squashing*: the problem of encoding an exponentially growing receptive field [1] in a fixed-size node embedding dimension. Topping et al. [13] have provided theoretical insights into this issue by identifying over-squashing with the exponential decrease in sensitivity of node representations to the input features on distant nodes as the number of layers increases. For example, a GCN model [8] computes the representation $\mathbf{h}_v^{(\ell)} \in \mathbb{R}^H$ of node $v$ in layer $\ell$ as the aggregation of previous-layer features in neighbouring nodes $v' \in \mathcal{N}(v)$, i. e.

$$\mathbf{h}_v^{(\ell)} = \mathrm{relu}\left(\sum_{v' \in \mathcal{N}(v)} \hat{\mathbf{A}}_{v,v'} \mathbf{W}^{(\ell)} \mathbf{h}_{v'}^{(\ell-1)}\right), \tag{1}$$

with $\hat{\mathbf{A}}$ as the normalized graph adjacency matrix and input node features $\mathbf{x}_v \in \mathbb{R}^X$ in layer $\ell = 1$. The sensitivity of $\mathbf{h}_v^{(\ell)}$ to the input $\mathbf{x}_{v'}$, assuming that there exists a $\ell$-path between nodes $v$ and $v'$, is upper bounded by

$$\left\|\frac{\partial \mathbf{h}_v^{(\ell)}}{\partial \mathbf{x}_{v'}}\right\| \leq \underbrace{\prod_{l=1}^{\ell} \|\mathbf{W}^{(l)}\|}_{\text{layers' Lipschitz constants}} (\hat{\mathbf{A}}^\ell)_{v,v'}. \tag{2}$$

D. Tortorella et al., Leave Graphs Alone: Addressing Over-Squashing without Rewiring (Extended Abstract). Presented at the First Learning on Graphs Conference (LoG 2022), Virtual Event, December 9–12, 2022.

Topping et al. [13] have further investigated the connection of over-squashing — as measured by the Jacobian of node representations in (2) — with the graph topology via the term $(\hat{\mathbf{A}}^\ell)_{v,v'}$, and have identified in negative local graph curvature the cause of 'bottlenecks' in message propagation. In order to remove these bottlenecks, they have proposed rewiring the input graph, i.e. altering the original set of edges as a preprocessing step, via *Stochastic Discrete Ricci Flow* (SDRF). This method works by iteratively adding an edge to support the most negatively-curved edge while removing the most positively-curved one according to the *balanced Forman curvature* [13], until convergence or a maximum number of iterations is reached. This rewiring approach can be contrasted to e.g. *Graph Diffusion Convolution* (DIGL) [14], which aims to address the problem of noisy edges in the input graph by altering the connectivity according to a generalized graph diffusion process, such as personalized PageRank (PPR). Since DIGL has a smoothing effect on the graph adjacency — by promoting connectivity between nodes that are a short diffusion distance —, it may be more suitable for tasks that present a high degree of homophily [13], i.e. graphs with an high ratio of intra-class edges [11].

In our opinion, equation (2) instead suggests a different method of addressing the exponentially vanishing sensitivity in deeper layers, by acting on the layers' Lipschitz constants $\|\mathbf{W}^{(l)}\|$. In the next section, we present a model for computing node embeddings in which Lipschitz constants can be explicitly chosen as part of the hyper-parameter selection. This will enable an experimental comparison between the two approaches in section 3.

## 2 Reservoir Computing for Graphs

Reservoir computing [15–17] is a paradigm for the efficient design of recurrent neural networks (RNNs). Input data is encoded by a randomly initialized reservoir, while only the readout layer for downstream task predictions requires training. Reservoir computing models, in particular Echo State Networks (ESNs) [18], have been studied in order to obtain insights into the architectural bias of RNNs [19, 20].

Graph Echo State Networks (GESNs) have been introduced by Gallicchio and Micheli [21], extending the reservoir computing paradigm to graph-structured data. GESNs have already demonstrated their effectiveness in graph-level classification tasks [22], and more recently in node-level classification tasks [23], in particular when the underlying graphs present low homophily. Node embeddings are recursively computed by the non-linear dynamical system

$$\mathbf{h}_v^{(k)} = \tanh\left(\mathbf{W}_{\text{in}}\,\mathbf{x}_v + \sum_{v'\in\mathcal{N}(v)}\hat{\mathbf{W}}\,\mathbf{h}_{v'}^{(k-1)}\right), \quad \mathbf{h}_v^{(0)} = \mathbf{0}, \tag{3}$$

where $\mathbf{W}_{\text{in}} \in \mathbb{R}^{H\times X}$ and $\hat{\mathbf{W}} \in \mathbb{R}^{H\times H}$ are the input-to-reservoir and the recurrent weights, respectively, for a reservoir with $H$ units (input bias is omitted). Equation (3) is iterated over $k$ until the system state converges to fixed point $\mathbf{h}_v^{(\infty)}$, which is used as the embedding. For node classification tasks, a linear readout is applied to node embeddings $\mathbf{y}_v = \mathbf{W}_{\text{out}}\,\mathbf{h}_v^{(\infty)} + \mathbf{b}_{\text{out}}$, where the weights $\mathbf{W}_{\text{out}} \in \mathbb{R}^{C\times H}, \mathbf{b}_{\text{out}} \in \mathbb{R}^C$ are trained by ridge regression on one-hot encodings of target classes $y_v$. The existence of a fixed point is guaranteed by the Graph Embedding Stability (GES) property [22], which also guarantees independence from the system's initial state $\mathbf{h}_v^{(0)}$. A sufficient condition for the GES property is requiring that the transition function defined in (3) to be contractive, i.e. to have Lipschitz constant $\|\hat{\mathbf{W}}\|\,\|\mathbf{A}\| < 1$. In standard reservoir computing practice, however, the recurrent weights are initialized according to a necessary condition [24] for the GES property, which is $\rho(\hat{\mathbf{W}}) < 1/\alpha$, where $\rho(\cdot)$ denotes the spectral radius of a matrix, i.e. its largest absolute eigenvalue, and $\alpha = \rho(\mathbf{A})$ is the graph spectral radius. This condition provides the best estimate of the system bifurcation point, i.e. the threshold beyond which (3) becomes asymptotically unstable [24]. Reservoir weights are randomly initialized from a uniform distribution in $[-1, 1]$, and then rescaled to the desired input scaling and reservoir spectral radius, without requiring any training.

Let us now consider a GESN where the number of iterations of (3) is fixed to a constant $K$. In this case, the $K$ iterations of the state transition function (3) can be interpreted as equivalent to $\ell = K$ graph convolution layers with weights shared among layers and input skip connections. In such a network, we are able to control how large the layers' Lipschitz constant is by increasing $\rho(\hat{\mathbf{W}})$, since the spectral radius is a lower bound for the spectral norm [25], i.e. $\|\hat{\mathbf{W}}\| \geq \rho(\hat{\mathbf{W}})$. This should allow us to contrast the exponentially vanishing sensitivity in (2) caused by topological bottlenecks in the

**Table 1:** Average test accuracy with $95\%$ confidence intervals (best results in bold). Except for GESN, the other results are reported from [13].

|  | Cornell | Texas | Wisconsin | Chameleon | Squirrel | Actor |
|---|---|---|---|---|---|---|
| None | $52.69_{\pm 0.21}$ | $61.19_{\pm 0.49}$ | $54.60_{\pm 0.86}$ | $41.80_{\pm 0.41}$ | $39.83_{\pm 0.14}$ | $28.70_{\pm 0.09}$ |
| Undirected | $53.20_{\pm 0.53}$ | $63.38_{\pm 0.87}$ | $51.37_{\pm 1.15}$ | $42.63_{\pm 0.30}$ | $40.77_{\pm 0.16}$ | $28.10_{\pm 0.11}$ |
| +FA | $58.29_{\pm 0.49}$ | $64.82_{\pm 0.29}$ | $55.48_{\pm 0.62}$ | $42.33_{\pm 0.17}$ | $40.74_{\pm 0.13}$ | $28.68_{\pm 0.16}$ |
| DIGL (PPR) | $58.26_{\pm 0.50}$ | $62.03_{\pm 0.43}$ | $49.53_{\pm 0.27}$ | $42.02_{\pm 0.13}$ | $34.38_{\pm 0.11}$ | $30.79_{\pm 0.10}$ |
| DIGL + Undir. | $59.54_{\pm 0.64}$ | $63.54_{\pm 0.38}$ | $52.23_{\pm 0.54}$ | $42.68_{\pm 0.12}$ | $33.36_{\pm 0.21}$ | $29.71_{\pm 0.11}$ |
| SDRF | $54.60_{\pm 0.39}$ | $64.46_{\pm 0.38}$ | $55.51_{\pm 0.27}$ | $43.75_{\pm 0.31}$ | $40.97_{\pm 0.14}$ | $29.70_{\pm 0.13}$ |
| SDRF + Undir. | $57.54_{\pm 0.34}$ | $70.35_{\pm 0.60}$ | $61.55_{\pm 0.86}$ | $44.46_{\pm 0.17}$ | $41.47_{\pm 0.21}$ | $29.85_{\pm 0.07}$ |
| GESN | $\mathbf{69.75}_{\pm 1.11}$ | $\mathbf{73.96}_{\pm 1.45}$ | $\mathbf{77.76}_{\pm 1.68}$ | $\mathbf{50.19}_{\pm 0.65}$ | $\mathbf{42.70}_{\pm 0.29}$ | $\mathbf{35.07}_{\pm 0.24}$ |

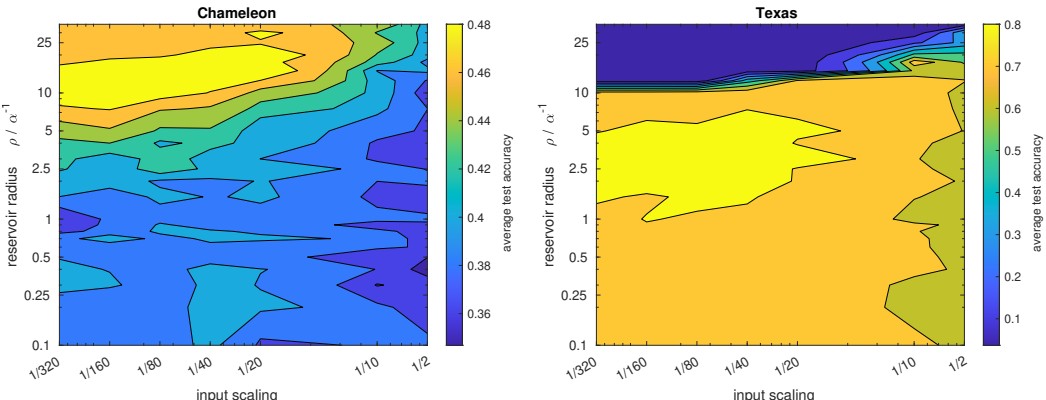

**Figure 1:** The effects of an adequately large reservoir radius $\rho$ (and thus of a large enough layer's Lipschitz constant, since $\|\hat{\mathbf{W}}\| \geq \rho$ [25]) on test accuracy for different input scaling factors on two of the six tasks.

factor $(\hat{\mathbf{A}}^{\ell})_{v,v'}$ with the contributions from the factor $\|\hat{\mathbf{W}}\|^{K}$, which is increasing with the number of iterations (unfolded recursive layers) if $\|\hat{\mathbf{W}}\| > 1$. Indeed, a preliminary work by Tortorella and Micheli [23] has empirically suggested that in tasks where the graph structure is relevant in the prediction, better node embeddings are computed well beyond the stability threshold.

## 3 Experiments and Discussion

In this section, we compare the accuracy of GESNs on six low-homophily node classification tasks against different rewiring mechanisms applied in conjunction with fully-trained GCNs. As Topping et al. [13] pointed out, avoiding over-squashing in order to capture long-range dependencies is often more relevant in low-homophily settings, since most nodes sharing the same labels are not neighbours. In our experiments we follow the same setting and training/validation/test splits of [13, 14], with tasks limited to the largest connected component of the original graphs, and report the average accuracy with $95\%$ confidence intervals on 1000 test bootstraps. As in [23], the hyper-parameters selected on the validation split for GESN are: the reservoir radius $\rho(\hat{\mathbf{W}})$, which controls how large the Lipschitz constant of (3) should be, in the range $[0.1/\alpha, 35/\alpha]$ (the range $\rho > 1/\alpha$ is obtained by grid search); the input scaling factor of $\mathbf{W}_{\text{in}}$ in the range $[\frac{1}{320}, 1]$; the number of units $H$ in the range $[2^4, 2^{12}]$; and the readout regularization for the ridge regression. The number of iterations is fixed at $K = 100$.

In Table 1 we compare the accuracy of GESN against the fully-adjacent (+FA) rewiring method by Alon and Yahav [12], the diffusion-based rewiring method DIGL (with PPR) by Gasteiger et al. [14], and the curvature-based graph rewiring method by Topping et al. [13] (for details on these models and their hyper-parameters, we refer to [13], where experimental results are taken from). We observe that GESNs beat the other models by a significant margin on all the six tasks. Indeed, DIGL and SDRF offer improvements over the baseline GCN of a few accuracy points on average, usually

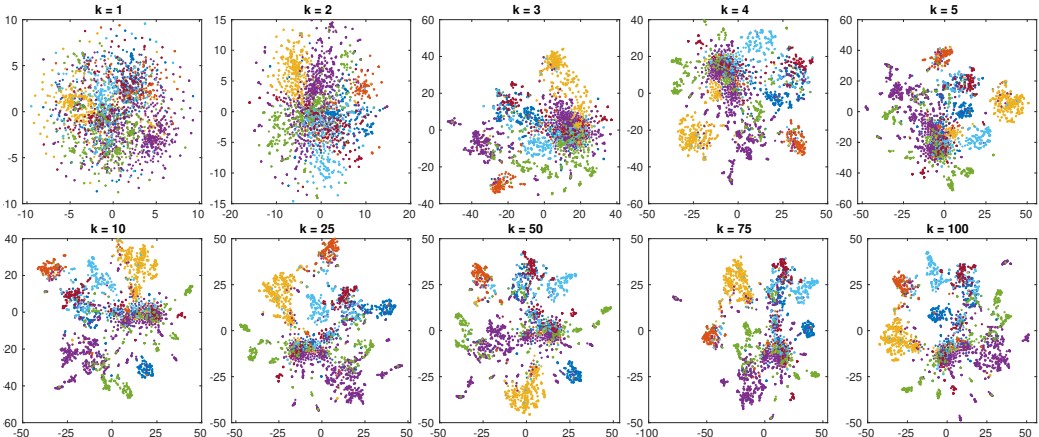

**Figure 2:** Node embeddings for the Cora graph at different iterations $k$ ($\rho = 6/\alpha$, 4096 units). Colours in the t-SNE plots represent different node classes, qualitatively showing how well separable are the node representations.

requiring also that the graph to be made undirected. In contrast, GESN improves up to $16\%$ over the best rewiring methods, and by 4-6 points on average. Notice also that rewiring algorithms, in particular SDRF, can be extremely costly and need careful tuning in model selection, in contrast to the efficiency of the reservoir computing approach, which ditches both the preprocessing of input graphs and the training of the node embedding function. Indeed, just the preprocessing step of SDRF can require computations ranging from the order of minutes to hours, while a complete model can be obtained with GESN in a few seconds' time on the same GPU.

Figure 1 shows the impact of reservoir radius $\rho$ and input scaling on test accuracy for Chameleon and Texas. An adequately large reservoir radius $\rho > 1/\alpha$, which in turn gives a large enough Lipschitz constant, is crucial in providing a significant gain in accuracy. Notice also that setting a proper input scaling is relevant, since it cannot be automatically adjusted by training as in GCNs via gradient descent. As a further insight, in Figure 2 we present the t-SNE plots of node embeddings of the Cora graph computed at different iterations of (3) with reservoir radius set at $\rho = 6/\alpha$. In GESNs, the iterations of the recursive transition function can be interpreted as equivalent to layers in deep message-passing graph networks where weights are shared among layers, in analogy with the unrolling in RNNs for sequences. We observe that instead of the collapse of node representations that has been shown in Li et al. [9] and subsequent works on the over-smoothing issue, node embeddings become more and more separable as the number of iterations increases. This observation, in conjunction with the accuracy results of Table 1 and of [23], suggests that the contractivity of the message-passing function, i.e. whether its Lipschitz constant is smaller or larger than 1, is the critical factor in addressing the degradation of accuracy in deep graph neural networks. Indeed, tuning the layer contractivity was implicitly done by Chen et al. [26] via a regularization term that favours larger pairwise distances of node representations as a mean to address the over-smoothing problem.

## 4 Conclusion

Motivated by the analysis of over-squashing via sensitivity to input features advanced by Topping et al. [13], we have proposed a different route to address this issue affecting the capability of deep graph neural networks to learn effective node representations. Instead of altering the input graph connectivity — as rewiring methods such as SDRF and DIGL propose —, we have shown that a model able to select the suitable Lipschitz constant for its graph convolution can achieve a significantly better accuracy on six node classification tasks with low homophily, even computing the node embeddings in a completely unsupervised and untrained fashion. Future work will involve investigating how the change in Lipschitz constant affects the organization of the node embedding space, and assessing the merit of transferring those results in fully-trained graph convolution models via a regularization term or via constraints on layers' weights.

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

## A  Comparison with node classification models

For the sake of completeness, in Table 2 we report accuracy of GESN and other node classification models on nine graphs with different degrees of homophily, following the experimental setting of Zhu et al. [11]. Notice that in this setting the whole graph of the task is retained, thus the results cannot be compared with those of Table 1, where graphs are restricted to the largest connected component following the setting of [13, 14]. The results show that GESN is effective on tasks with high homophily as well as on tasks with low homophily, thanks to the ability to tune the Lipschitz constant of (3).

**Table 2:** Node classification accuracy on low and high homophily graphs following the experimental setting of Zhu et al. [11]. Average accuracy and standard deviation for GESN is reported from [23], while other models are reported from [11]. Results within one standard deviation of the best accuracy are highlighted.

| | Texas | Wisconsin | Actor | Squirrel | Chameleon | Cornell | Citeseer | Pubmed | Cora |
|---|---|---|---|---|---|---|---|---|---|
| GraphSAGE | $82.4_{\pm6.1}$ | $81.2_{\pm5.6}$ | $34.2_{\pm1.0}$ | $41.6_{\pm0.7}$ | $58.7_{\pm1.7}$ | $75.9_{\pm5.0}$ | $76.0_{\pm1.3}$ | $88.5_{\pm0.5}$ | $86.9_{\pm1.0}$ |
| GAT | $58.4_{\pm4.5}$ | $55.3_{\pm8.7}$ | $26.3_{\pm1.7}$ | $30.6_{\pm2.1}$ | $54.7_{\pm1.9}$ | $58.9_{\pm3.3}$ | $75.5_{\pm1.7}$ | $84.7_{\pm0.4}$ | $82.7_{\pm1.8}$ |
| GCN | $59.5_{\pm5.3}$ | $59.8_{\pm7.0}$ | $30.3_{\pm0.8}$ | $36.9_{\pm1.3}$ | $59.8_{\pm2.6}$ | $57.0_{\pm4.7}$ | $76.7_{\pm1.6}$ | $87.4_{\pm0.7}$ | $87.3_{\pm1.3}$ |
| GCN+JK | $66.5_{\pm6.6}$ | $74.3_{\pm6.4}$ | $34.2_{\pm0.9}$ | $40.5_{\pm1.6}$ | $63.4_{\pm2.0}$ | $64.6_{\pm8.7}$ | $74.5_{\pm1.8}$ | $88.4_{\pm0.5}$ | $85.8_{\pm0.9}$ |
| GCN+Cheby | $77.3_{\pm4.1}$ | $79.4_{\pm4.5}$ | $34.1_{\pm1.1}$ | $43.9_{\pm1.6}$ | $55.2_{\pm2.8}$ | $74.3_{\pm7.5}$ | $75.8_{\pm1.5}$ | $88.7_{\pm0.6}$ | $86.8_{\pm1.0}$ |
| MixHop | $77.8_{\pm7.7}$ | $75.9_{\pm4.9}$ | $32.2_{\pm2.3}$ | $43.8_{\pm1.5}$ | $60.5_{\pm2.5}$ | $73.5_{\pm6.3}$ | $76.3_{\pm1.3}$ | $85.3_{\pm0.6}$ | $87.6_{\pm0.9}$ |
| H2GCN | $84.9_{\pm6.8}$ | $86.7_{\pm4.7}$ | $35.9_{\pm1.0}$ | $36.4_{\pm1.9}$ | $57.1_{\pm1.6}$ | $82.2_{\pm4.8}$ | $77.1_{\pm1.6}$ | $89.4_{\pm0.3}$ | $86.9_{\pm1.4}$ |
| MLP | $81.9_{\pm4.8}$ | $85.3_{\pm3.6}$ | $35.8_{\pm1.0}$ | $29.7_{\pm1.8}$ | $46.4_{\pm2.5}$ | $81.1_{\pm6.4}$ | $72.4_{\pm2.2}$ | $86.7_{\pm0.4}$ | $74.8_{\pm2.2}$ |
| GESN | $84.3_{\pm4.4}$ | $83.3_{\pm3.8}$ | $34.5_{\pm0.8}$ | $71.2_{\pm1.5}$ | $76.2_{\pm1.2}$ | $81.1_{\pm6.0}$ | $74.5_{\pm2.1}$ | $89.2_{\pm0.3}$ | $86.0_{\pm1.0}$ |

**Table 3:** Statistics for the tasks in Table 2.

| Task | Homophily | Nodes | Edges | Radius $\alpha$ | Features | Classes |
|---|---|---|---|---|---|---|
| Texas | 0.11 | 183 | 295 | 2.56 | 1,703 | 5 |
| Wisconsin | 0.21 | 251 | 466 | 2.88 | 1,703 | 5 |
| Actor | 0.22 | 7,600 | 26,752 | 9.99 | 932 | 5 |
| Squirrel | 0.22 | 5,201 | 198,493 | 138.60 | 2,089 | 5 |
| Chameleon | 0.23 | 2,277 | 31,421 | 61.90 | 2,089 | 5 |
| Cornell | 0.30 | 183 | 280 | 2.68 | 1,703 | 5 |
| Citeseer | 0.74 | 3,327 | 9,104 | 13.74 | 3,703 | 6 |
| Pubmed | 0.80 | 19,717 | 88,648 | 23.24 | 500 | 3 |
| Cora | 0.81 | 2,708 | 10,556 | 14.39 | 1,433 | 7 |

## B  Role of reservoir radius

In Figure 3, we show the impact of reservoir radius $\rho$ and input scaling factor on average test accuracy for the tasks in Appendix A, reaffirming the analysis of Tortorella and Micheli [23]. Chameleon and Squirrel (two tasks with low homophily) require an extremely large reservoir radius, while essentially ignoring the input features due to the extremely small input scaling factor. This suggests that having a large Lipschitz constant is beneficial for the extraction of relevant topological features from the graph. The other four low homophily tasks (Actor, Cornell, Texas, Wisconsin) seem to exploit more the information of node input labels instead of graph connectivity, by requiring reservoir radii within the stability threshold. Finally, the three high homophily tasks (Cora, Citeseer, Pubmed) achieve the best accuracy with a combination of moderately high spectral radius and input scaling relatively close to 1. Overall, what we have observed shows that GESN can be flexible enough to accommodate the two opposite task requirements thanks to the explicit tuning of both input scaling and reservoir radius in the model selection phase.

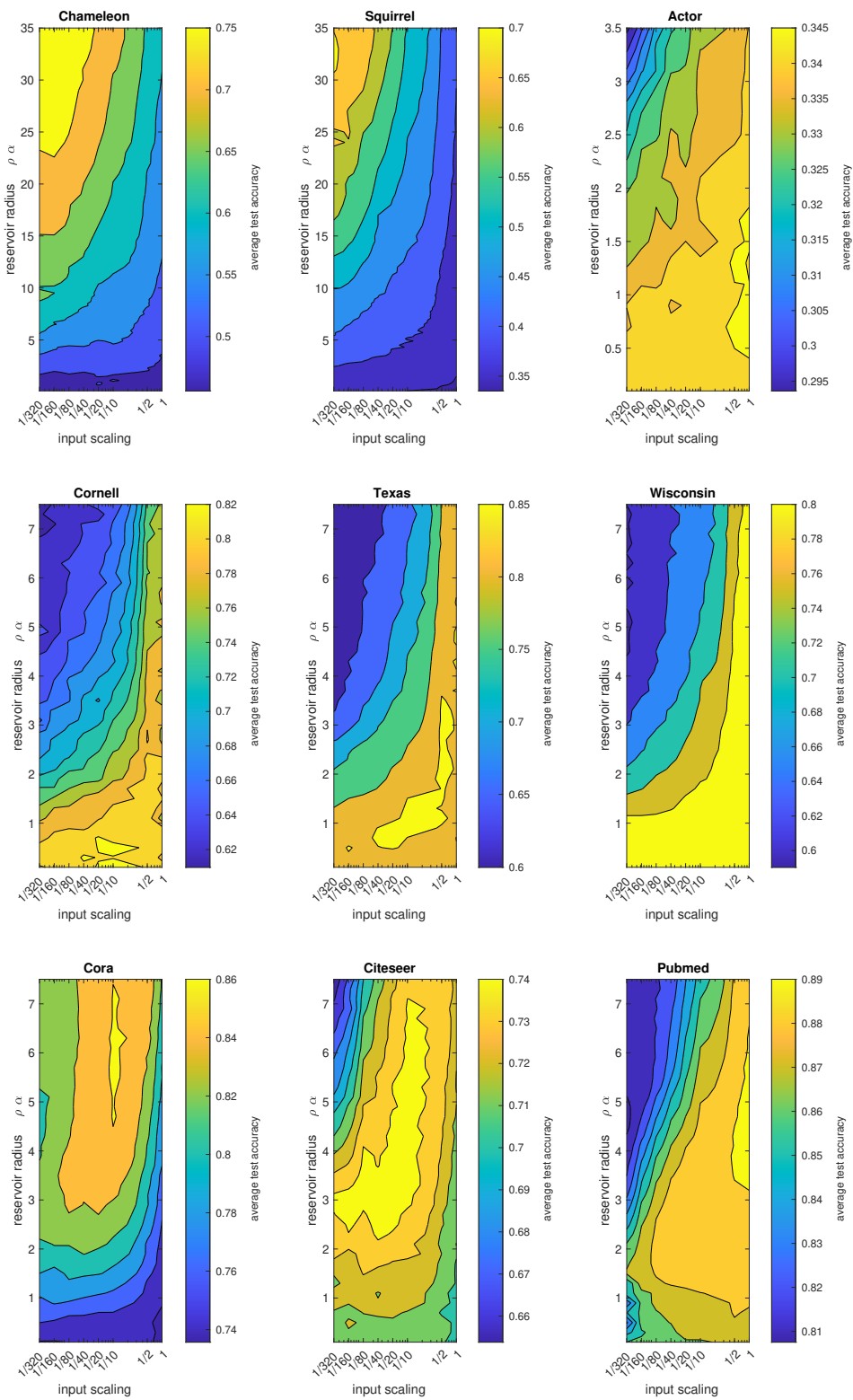

**Figure 3:** Impact of input scaling and reservoir radius on test accuracy (4096 units).

