# OpenReview forum: "Leave Graphs Alone: Addressing Over-Squashing without Rewiring"
_logconference.io/LOG/2022/Conference — LoG 2022 Poster_

### Official Review · Reviewer_UbKW · 2022-09-22

**Overall Score:** 6
**Confidence:** 4

**Review:**

**1. Summary of Contributions**

This paper considers the oversquashing phenomenon in message-passing graph neural networks (GNNs), which is that deep GNNs encode exponentially growing information into fixed-size vectors.

As a remedy, prior work has proposed to modify the graph structure, but this paper retains the original graph structure and proposes to modify the GNN model instead.

The proposed model, inspired by reservoir computing [1] and its recent explorations on graphs, trains node embeddings only at a final read-out stage (a key benefit of reservoir computing), and is more accurate, faster to train than prior work as demonstrated on real-world heterophilic datasets.

[1] Gouhei et al., Recent advances in physical reservoir computing: A review, In Neural Networks 2019.


\
**2.Strong Points and Weak Points**

**Strong Points**

\+ The high-level ideas are easy to follow.

\+ The motivation of retaining the input graph structure but modifying the model instead for oversquashing is strong and can potentially be useful for processing graphs such as molecules where input graphs are very important.

\+ The proposed model naturally comes with the key benefits of reservoir computing, viz., ability to train node embeddings in an unsupervised manner, or at a final read-out stage, and faster training than prior work.

**Weak Points**
\- A short discussion (lines 50-54)  describes how the proposed model could better overcome oversquashing (than prior work) but connections between reservoir computing and oversquashing remain largely unclear.

\- Experiments are conducted on heterophilic datasets but differences from and comparisons with an existing cited work [Beyond homophily with graph echo state networks, In ESANN'22] are missing.

\- The datasets are small and faster training times could be better appreciated on larger non-homophilous datasets [Large Scale Learning on Non-Homophilous Graphs: New Benchmarks and Strong Simple Methods, In NeurIPS'21].

\
**3.Recommendation**

Weak reject.

While there exist potential benefits of the method, the connections between the proposed method (reservoir computing) and the problem to address (oversquashing) remain unclear and the benefits of the method could be better appreciated on larger datasets with comparison of similar missing methods.


**4. Supporting Arguments for Recommendation**

While experiments on real-world heterophilic datasets are necessary, experiments on synthetic datasets can reveal insights into the proposed method. A relevant synthetic dataset, which was first used in prior work to analyse oversquashing, is The Tree-NeighboursMatch problem [2]. The authors are suggested to compare the proposed method with GNNs equipped with fully-adjacent layers as done previously [2] on Tree-NeighboursMatch with increasing tree depths.

[2] On the Bottleneck of Graph Neural Networks and its Practical Implications, In ICLR'21.

\
**5. Questions to Authors**
1. How is the proposed model in this paper different from the model in [Beyond homophily with graph echo state networks, In ESANN'22]?
2. The authors describe on lines 52 and 53 that the Lipshitz constants can be explicitly chosen as part of the hyper-parameter selection. How does this help in mitigating oversquashing? Lines 82-90 seem to have an explanation but the details are referred to a cited work [Beyond homophily with graph echo state networks, In ESANN'22].
3. On lines 95-97, the authors give the ranges for selection of some of the key hyperparameters but the significance of the particular numbers chosen is unclear. More specifically, what are the significances of the numbers 0.1 and 35 in the range [0.1, 35) for the reservoir radius? 1/320 for input scaling?


**6.Additional Questions/Feedback to Authors**

* Line 8: singnificantly -> significantly

* What is \rho\alpha on Line 112?

* How does the model compare with simplified graph convolution (SGC) which, for node classification, is essentially a logisitic regression-based model? SGC is similar to reservoir computing in the sense of having a final read-out stage after performing feature propagations over the graph.  A recent work proposes an adaptation of SGC to heterophilic datasets [Simplified Graph Convolution with Heterophily, In NeurIPS'22].


**7. Type of Paper**

4-page track

\
**Update**

I have read all the reviews and their author responses. The major concerns regarding clarity on reservoir computing and relation with prior work seem to have been addressed. I have changed my rating from weak reject to weak accept.

An important concern that remains even after the rebuttal is the missing experiments on relevant data (e.g., TreeNeighbours Match and Large-scale non-homophilous data).

---

### Official Review · Reviewer_ETHm · 2022-10-14

**Overall Score:** 6
**Confidence:** 3

**Review:**

This paper proposes to revisit the Graph Echo State Network (GESN) architecture to address the over-squashing problem of GNNs.

I am only listing the rating 6 because of the limited contribution of the work. Essentially, it consists of (i) observation that the over-smoothing related to the Lipschitz constant of GNN, and we can directly influence that, and (ii) running an experimental evaluation of such model and comparison with modern competitors. In my personal opinion, these contributions are valuable enough to be presented.

The paper is easy to follow and well-motivated, the observation that is the core idea of the paper is very nice. My only suggestion would be to somehow demonstrate the Lipschitz constant in the similar way to Figure 2 of the Appendix. I believe that the relationship between homophily and over-squashing is not very well explored, but this is easily forgivable for a 4-page non-archival paper.

As an aside, I like the title of the paper. :-)

---

### Official Review · Reviewer_YykY · 2022-10-17

**Overall Score:** 5
**Confidence:** 3

**Review:**

Summary:

In this paper, the author(s) try to use Graph Echo State Networks to address ‘over-squashing’ problem. The method is largely inspired by Topping et al.’s work Understanding over-squashing and bottlenecks on graphs via curvature. Instead of rewiring the input graph, they use GESN to control layers’ Lipshitz constants.

Strengths:

1.	Compared to the previous work, there is a big improvement in terms of results.

Weaknesses:

1.	The presentation of the model used in the article is somewhat vague, and even some parameters in it are not presented.

2.	This article argues that the earlier work is more appropriate for the homophily graph scenario, whereas this article is more concerned with the scenario in the heterophily graph. However, the article does not address the necessity of using such a model for the heterophily graph scenario.

3.	When the authors evaluate the performance of the model, only a fairly limited amount of work is compared with it in the body of the article. And in the results presented in the appendix, there is not always a significant improvement in the results of the models used in the article in the low homophily case. There are also some discrepancies between the model effects in the appendix and those mentioned in the main text of the article, which should be explained.

---

### Official Review · Reviewer_M6jW · 2022-10-22

**Overall Score:** 6
**Confidence:** 3

**Review:**

Summary:

This is a 4-page paper. This paper addresses the over-squashing issue in GNNs using the tool from [[reservior computing]]. In contrast to the conventional remedy of graph rewiring, the authors propose to use Graph Echo State Networks (GESN, Gallicchio and Micheli) without modifying the graph connectivity. In the experiments over six low-homophily benchmarks, GESNs are shown to performan significantly better, even though the node embeddings are completely untrained (and clearly unsupervised).

Reason for score:

Overall, I vote for accepting. I like the simplicity and effectiveness of the idea. Unlike graph rewiring, which might require additional preprocessing, GESNs do not rely on heuristics to rewire the graph connectivity beforehand. Even surprsingly, the node embeddings are untrained (the reservoir is fixed) and unsupervised. My main concern would be about the novelty and the clarity on the related work while I understand that there is only so much you can fit into 4 pages.

Pros:

- The idea of tackling over-squashing with reservior computing is interesting to me. Essentially, some part of the model parameters are frozen (randomly intialised) and the node embeddings are computed by repeatedly forward-passing through these frozen parameters. The final node embeddings (the fixed point solution)  are then fed to a learnable readout function.
- The experiments over six benchmarks (low-homophily, node classification) show great improvement of GESN over the baselines. The improvement seems impressive.


Cons:

I understand that 4 pages might not allow much content. But it would be great if the author can address the following questions:
- In terms of novelty, I wonder what is the main difference between this work and the GESN paper. Is the novelty here mainly on applying GESN to low-homophily datasets?
- In terms of related work, there are some general machine learning works, which also tried to recursively compute the representation using the same set of frozen weights, i.e. identical layers repeated for K times. For example, in NLP, [ALBERT](https://arxiv.org/abs/1909.11942) and [ReFactorGNN](https://arxiv.org/abs/2207.09980) both use identical layers to save computation, capture global structures, and for regularisation purposes. I wonder if your work share the same spirit. In general, is there any connection between identical layers in neural networks and reservoir computing? If not, it would be worthwhile to point it out.
- In terms of baseline in Table 1,  I wonder if factorisation-based models are a sensible baseline. They implicitly contain infinite-layer message-passing as shown by  [ReFactorGNN](https://arxiv.org/abs/2207.09980) and should be suitable for capturing long-range information.
- Minor. Line 79, what is bifurcation point?


Questions:

Please address the cons above.

Typos:
- 133 o -> or?

---

### Meta-Review · Area_Chair_MYd9 · 2022-11-15

**Confidence:** 4
**Recommendation:** Accept

**Meta Review:**

The reviewers agree that the paper should be accepted.

## Strengths
(S1) Reviewers appreciated strong experimental results over baselines.
(S2) Reviewers felt that this paper introduced novel methods.

## Weaknesses
(W1) The reviewers found that the paper would profit from evaluation on large-scale datasets.

I recommend acceptance of this manuscript, with the following weighting of importance: (S2) > (S2) > (W1).

---

### Decision · Program_Chairs · 2022-11-23

Accept (Poster)